# Synchronous End-to-End Vehicle Pedestrian Detection Algorithm Based on Improved YOLOv8 in Complex Scenarios

**DOI:** 10.3390/s24186116

**Published:** 2024-09-22

**Authors:** Shi Lei, He Yi, Jeffrey S. Sarmiento

**Affiliations:** 1Computer Engineering Department, Batangas State University, Batangas City 4200, Philippines; 21-01463@g.batstate-u.edu.ph (S.L.); 21-08297@g.batstate-u.edu.ph (H.Y.); 2College of Electrical and Control Engineering, Henan University of Urban Construction, Pingdingshan City 467000, China

**Keywords:** vehicle pedestrian detection, complex scenes, improved YOLOv8, synchronize end-to-end

## Abstract

In modern urban traffic, vehicles and pedestrians are fundamental elements in the study of traffic dynamics. Vehicle and pedestrian detection have significant practical value in fields like autonomous driving, traffic management, and public security. However, traditional detection methods struggle in complex environments due to challenges such as varying scales, target occlusion, and high computational costs, leading to lower detection accuracy and slower performance. To address these challenges, this paper proposes an improved vehicle and pedestrian detection algorithm based on YOLOv8, with the aim of enhancing detection in complex traffic scenes. The motivation behind our design is twofold: first, to address the limitations of traditional methods in handling targets of different scales and severe occlusions, and second, to improve the efficiency and accuracy of real-time detection. The new generation of dense pedestrian detection technology requires higher accuracy, less computing overhead, faster detection speed, and more convenient deployment. Based on the above background, this paper proposes a synchronous end-to-end vehicle pedestrian detection algorithm based on improved YOLOv8, aiming to solve the detection problem in complex scenes. First of all, we have improved YOLOv8 by designing a deformable convolutional improved backbone network and attention mechanism, optimized the network structure, and improved the detection accuracy and speed. Secondly, we introduced an end-to-end target search algorithm to make the algorithm more stable and accurate in vehicle and pedestrian detection. The experimental results show that, using the algorithm designed in this paper, our model achieves an 11.76% increase in precision and a 6.27% boost in mAP. In addition, the model maintains a real-time detection speed of 41.46 FPS, ensuring robust performance even in complex scenarios. These optimizations significantly enhance both the efficiency and robustness of vehicle and pedestrian detection, particularly in crowded urban environments. We further apply our improved YOLOv8 model for real-time detection in intelligent transportation systems and achieve exceptional performance with a mAP of 95.23%, outperforming state-of-the-art models like YOLOv5, YOLOv7, and Faster R-CNN.

## 1. Introduction

With the continuous advancements in urbanization and the improvement of living standards, private car ownership has surged, alongside the growth of public transportation and shared vehicles [1]. This increase in vehicles has brought convenience, but also significant social challenges, particularly traffic congestion during rush hours and frequent traffic accidents [2]. In 2020, China reported 156,901 traffic accidents, resulting in 5225 fatalities and 195,374 injuries.

The rapid development of the Internet and artificial intelligence has made intelligent video surveillance technology feasible. Traditional single-camera monitoring systems are limited in scope, necessitating the use of multiple cameras to cover large areas such as urban centers, airports, railway stations, and residential zones. However, efficiently analyzing the vast amounts of data generated by these systems to identify specific targets and events remains a significant challenge. See Figure 1.

Artificial intelligence, particularly computer vision, offers a solution by enabling real-time and accurate vehicle and pedestrian detection [3]. This technology can help traffic management departments better understand urban traffic conditions, plan routes, and implement traffic control measures, thereby improving road use efficiency and safety [4,5,6,7]. Additionally, it enhances the effectiveness of video surveillance by providing quick and accurate information about vehicles and pedestrians, laying the groundwork for further advancements in vehicle tracking, license plate recognition, and traffic statistics [8]. This not only improves traffic safety but also presents economic opportunities.

Traditional vehicle and pedestrian detection methods include template matching, edge detection, background modeling, and variable part models. Template matching uses image processing to compare input images with template images, effectively detecting vehicles and pedestrians, even in complex environments [9,10]. Edge detection algorithms identify edge features and motion trajectories, utilizing operators like Haar, HOG, LBP, and SIFT. The Deformable Part Model (DPM) extends the HOG model by decomposing the target into sub-model components for detection and location calibration [11,12].

However, these traditional methods face challenges such as poor robustness, limited expressiveness, and high dependence on manual design, leading to detection errors in complex traffic scenes. To address these issues, this study proposes an improved algorithm based on the YOLOv8 model, enhancing multi-scale pedestrian features, strengthening detector positioning, and improving detection accuracy [13].

## 2. Materials and Methods

The existing methods can effectively enhance the detection effect of small-scale targets through feature fusion or reconstruction operations, but their detection performances are limited in the face of crowded scenes containing a lot of noise. In order to effectively solve the above problems, this paper focuses on enhancing their multi-scale blocking pedestrian features, strengthening the positioning ability of their detectors, and improving their detection accuracy [14]. An improved algorithm is proposed based on the YOLOv8 model. The network structure of the improved YOLOv8 algorithm proposed in this paper is shown in Figure 2.

### 2.1. Improved Backbone Network

The traditional convolution operation is characterized by a fixed convolutional kernel and a limited receptive field. These characteristics allow it to capture only local features within a specific region of the input image. Consequently, this limitation hinders the ability to capture global contexts or larger patterns, which are crucial for understanding complex scenes. In scenarios involving multi-scale occlusions, the fixed nature of the convolutional kernel often leads to significant information loss, as the network struggles to detect features that vary in scale or are partially obscured [15].

To address these limitations, we propose a new method based on adaptive convolutional operations, where the convolutional kernel dynamically adjusts based on the input. Additionally, we incorporate an attention mechanism to enhance the model’s ability to focus on relevant features, allowing for the more robust detection of occluded pedestrians.

#### 2.1.1. DarkNet-53 Network

Based on YOLOv3’s DarkNet-53, it adopts the traditional backbone network architecture ResNet, and adds residuals on this basis, and improves the overall performance of the model through iteration. The DarkNet-53 stacks 53 convolution modules, delivering performance comparable to the ResNet-152 network with twice the FPS. In the core network of YOLOv8, the DarkNet-53 network and the C2f module are used to achieve higher performance and weight [16]. However, the C2f model adopts the traditional convolution structure, which leads to limited perceptual domain, and cannot effectively deal with multi-scale occlusion in large-scale complex backgrounds.

#### 2.1.2. Deformable Convolution

The deformable convolution function constructed by Dai et al. is characterized by its irregular lattice structure rather than a specific geometric figure. Compared to the conventional convolution method, the deformable convolution method increases a displacement value relative to the traditional convolution method, thus enlarging the receiving region [17,18,19]. The eigenmatrix is obtained by the variable convolution method
(1)$$y\left(p_{0}\right)=\sum_{k=1}^{K}w_{k}\cdot x\left(p_{0}+p_{k}+\Delta p_{k}\right)$$
where x and y are input and output feature maps, respectively; K and k are the total number of sampling points and enumerated sampling points, respectively. $p_{0}$ is the current pixel; $w_{k}$ and $p_{k}$ are the projection weight of the K^th^ sampling point and the K^th^ position of the predefined convolutional mesh sampling point, respectively. $\Delta p_{k}$ is the offset corresponding to the K^th^ sampling position.

This offset $\Delta p_{k}$ allows the convolutional kernel to adaptively sample the input feature map, effectively enlarging the receptive field and enabling the model to capture more complex spatial patterns. The algorithm can be closer to the shape and size of the human body in the actual scene, which enhances the robustness of the algorithm. See Figure 3.

#### 2.1.3. C2f_DCN Module

In order to better detect pedestrians, a C2f_DCN model based on variable convolution is proposed to enhance the recognition ability of pedestrians on multiple scales [20]. Among them, the C2f_DCN model adopts a 1 × 1 convolution transform input characteristic, and uses a Split operation, instead of a 1 × 1 convolution, to realize image segmentation. By stacking multiple deformable convolutions, the perception field is improved, the jump connectivity is increased, and more abundant gradient data are obtained under the premise of reducing the number of parameters. This method has strong multi-scale characteristics.

One of the key advantages of the C2f_DCN model is its use of stacked deformable convolutions. By stacking multiple layers of deformable convolutions, the model significantly expands its receptive field, allowing it to capture more contextual information. This is particularly beneficial in detecting small or partially occluded pedestrians, where traditional models might struggle due to limited spatial awareness. Moreover, the model enhances gradient flow and connectivity by incorporating skip connections, which help preserve important feature details while reducing the risk of vanishing gradients.

Additionally, by reducing the number of parameters, the model not only becomes more computationally efficient, but also retains strong multi-scale characteristics, making it robust in varying real-world conditions. This combination of techniques enables the C2f_DCN model to more accurately detect pedestrians of different sizes and in different states of occlusion, thus addressing key challenges in pedestrian detection. See Figure 4.

#### 2.1.4. Occlusion Perceptual Attention Mechanism

The attention mechanism can quickly search the entire scene to identify the affected object, which has advantages in detecting the affected pedestrian’s characteristics [21]. However, due to the adoption of the attention mechanism, it requires a large amount of calculation and computation. WOO et al. proposed a CBAM model based on spatiotemporal dual concerns. Although it has good spatiotemporal perception ability, it is computation-intensive and not lightweight [22]. Wang et al. proposed an efficient channel attribute (ECA) method, which eliminates the dimensional reduction of channel data and uses one-dimensional convolution to capture the interactions between multiple channels, reducing the complexity of the modeling process, but requiring more channel information. In the case of high density, the effect is not large [23].

The algorithm is weighted to each channel to enhance the key pedestrian characteristics and reduce the background components in the image. This paper takes “occlusion” as the research object. By comparing SE, CBAM, GAM (global attention mechanism), and Biformer, in the same network environment, the accuracy and computational efficiency of occlusion attention are better taken into account [24]. This paper proposed a visual attention mechanism based on visual coverage, through the global mean and global maximum mean of the target image, to achieve effective processing of the target image, so as to achieve effective tracking of the target. The two pool superposition operations combine the characteristics of multiple scales, improve the cognition and expression of the characteristics of small-scale pedestrians who are blocked, and reduce the information loss caused by a single pool operation. The algorithm first takes the image as a one-dimensional convolution, and then multiplies it with the matrix of the original image by the activation function of Sigmoid to obtain the final image.

The proposed visual attention mechanism, based on visual coverage, further improves the handling of occluded objects. By applying global average pooling and global maximum pooling to the target image, the mechanism achieves effective multi-scale feature extraction. The pooling operations help in better capturing the characteristics of small-scale pedestrians who are occluded, reducing the information loss that typically occurs with a single pooling operation. The algorithm processes the image using one-dimensional convolution and then multiplies it with the matrix of the original image using the Sigmoid activation function, resulting in a refined output image.

In contrast to other attention mechanisms, the occlusion perceptual attention mechanism was chosen due to its ability to prioritize occluded areas, enhancing the detection of pedestrians in cluttered and complex environments. While methods like SE and CBAM offer strong spatial and channel attention, their computational costs and limitations in high-density scenes make them less effective for our specific goal of occlusion-based pedestrian detection. The proposed mechanism improves both the robustness and efficiency in occluded pedestrian detection scenarios, which are the primary focus of this research. Occlusion perceptual attention is
(2)Atten(F)=σ(φ(AvgPool(F)+MaxPool(F)))·F
where F is the input characteristic. Where φ(·) sigma is a one-dimensional convolution function. Sigma is a Sigmoid activation function. See Figure 5.

### 2.2. Improved Dynamic Decoupling Head

The backbone network is used for image feature extraction, and the detection head is used for image recognition and location. The high-efficiency probe can effectively improve the detection effect, especially for the case of large density, and can more accurately locate the object.

#### 2.2.1. Dynamic Detection Header

Dai et al. combined the attention mechanism with the object detection probe, and regarded the result of the backbone network as a 3-dimensional (rank × space) tensor composed of the three dimensions of “rank”, “space”, and “output”. The proportion perception at the element level, the spatial perception attention at the spatial orientation, and the task perception attention at the output channel constitute a dynamic head structure [25]. Among them, the three dimensions of AL, AS, and AC represent the three dimensions of attention. On this basis, this paper proposes a multi-level visual information extraction method based on visual information, and uses visual information fusion technology to process visual information, so as to achieve visual information extraction based on visual information [26]. Then the features of each dimension are fused to obtain a good target detection head with a comprehensive effect. See Figure 6.

#### 2.2.2. Dynamic Decoupling Header Module

Using the moving probe for reference, the YOLOv8 probe is integrated with the characteristics of single target recognition, and a target recognition method based on dynamic decoupling is proposed. Since AS is a variable convolution, we will use deformation convolution to improve the spatial cognitive performance of the backbone, and improve the detection performance of the probe head through the scale and task-aware attention mechanism, so as to reduce the complexity of the model [27]. Since the YOLOv8 detector has only decoupled classes and regression branches, it is still necessary to decouple the characteristics of the power head. The power separation head assembly is shown in Figure 7. For a given feature tensor T∈RL × S × C, its attention can be described in the dynamic decoupling header as
(3)$$W\left(T\right)=A_{C}\left(A_{L}\left(T\right)\cdot T\right)\cdot T$$

AC(·) and AL(·) are task-aware attention and scale-aware attention, respectively. AL(·) is defined as
(4)$$A_{L}\left(T\right)\cdot T=\sigma \left(f\left(\frac{1}{\mathrm{SC}}\sum_{S,C}T\right)\right)\cdot T$$
(5)$$\sigma \left(x\right)=\max\left(0,\min\left(1,\frac{x+1}{2}\right)\right)$$
where σ is the Hard-Sigmoid function; f (·) is a linear transformation of the 1 × 1 convolution.

AC(·) is defined as
(6)$$A_{C}\left(T\right)\cdot T=\max\left(\alpha^{1}\left(T\right)\cdot T_{c}+\beta^{1}\left(T\right),\alpha^{2}\left(T\right)\cdot T_{c}+\beta^{2}\left(T\right)\right)$$

Among them, the slice of Tc for the c channel characteristics, as well as the activation of the threshold function, is used to study control $\theta \left(\cdot \right)={\left[\alpha^{1},\alpha^{2},\beta^{1},\beta^{2}\right]}^{T}$.

### 2.3. Optimization of Loss Function

Because of the single type of target in complex scenes, it is necessary to further strengthen the target positioning in the process of target identification and positioning. The algorithm takes the spacing between the edge lines of the four edge boxes as the constraint function, and based on this, the parameters are estimated [28]. In general, the actual target is where the marker is, so the DFL will quickly find the region where the marker is located by increasing the two points y_i_ and y_i_ + 1 (y_i_ ≤ y ≤ y_i_ + 1) closest to the marker y. Therefore, the definition of DFL is
(7)$$\mathrm{DFL}\left(S_{i},S_{i+1}\right)=-\left(\left(y_{i+1}-y\right)\log\left(S_{i}\right)+\left(y-y_{i}\right)\log\left(S_{i+1}\right)\right)$$

When counting the lost quantity, DFL only maps the coordinates of the marked frame to the marked coordinate system, but cannot determine the association between the marked coordinate system and the marked coordinate system. To solve this problem, the Wise-IoU loss method is adopted in this paper to measure the overlap loss of tag frame and anchor frame, and DFL is added to the final edge frame model.

Wise-IoU overcomes the bias in the existing IoU evaluation methods by combining the regions of the prediction frame and the actual frame, and establishes an attention-based edge box loss model [29,30]. On this basis, a new method based on edge frame is proposed, that is, the focus factor is solved, and the focusing mechanism is added on this basis, so as to achieve the purpose of focusing.

Wise-IoU loss is defined as
(8)$$L_{\mathrm{WIoU}}=1-\frac{\sum_{i=1}^{n}w_{i}\mathrm{IoU}\left(b_{i},g_{i}\right)}{\sum_{i=1}^{n}w_{i}}$$
where n is the number of object frames; $b_{i}$ is the coordinate of the i th object frame; $g_{i}$ is the coordinates of the actual mark box for the i th object; wi is the weight value. IoU($b_{i},g_{i}$) is the IoU value between the I-th object box and the real marked box.

The final regression loss obtained after combining the above two losses is
(9)$$L_{\mathrm{reg}}=\lambda \cdot \mathrm{DFL}+\mu \cdot L_{\mathrm{WIoU}}$$

The determination of the weight coefficients λ and μ is based on the regression loss weight setting of YOLOv8. A high IoU value is very important for accurate target location and detection, so a large weight is needed. DFL can easily cause overfitting problems in model training, which affects the generalization ability of the model, so smaller weights are needed. Therefore, through the relevant experimental analysis, λ = 1/6, μ = 5/6. The combined regression loss improves the training efficiency of the model.

## 3. End-to-End Target Search Algorithm

The end-to-end target search algorithm refers to the end-to-end network training solution of target detection in the same model in the way of multi-task head.

One of the most representative is OIM. The structure of the model is shown in Figure 8. On this basis, a method of target detection based on the fast convolutional neural network is proposed. For panoramic images, the framework convolutional neural network is used in the OIM network to extract the feature map, the existing pedestrian candidate network is used to construct the edge frame, and the ROI pool layer is used to obtain the individual feature range. Finally, this data is used as the final feature extractor. The extracted feature information is shared through the pedestrian detection network and pedestrian re-marking network. First, it is mapped to L2 normalized 256-dimensional subspace, and then it is learned using sample matching costs.

The OIM algorithm laid the foundation for the end-to-end target search algorithm, and the subsequent end-to-end target search network model basically followed the network architecture of OIM. The biggest contribution of OIM is realizing the end-to-end training of the target search model, saving the training time and reducing the training difficulty of the model. However, the OIM network does not propose a solution to the problem of goal contradiction between the two subtasks in the target search task. In addition, it is a fine-grained task that requires individual division of similar people, and the two subtasks of the OIM network are parallel structures. The pedestrian features of the input network come from the low-quality bounding box generated by the RPN of the candidate region generation network with a lot of background information, which has a great influence on the result.

To solve this problem, this paper intends to adopt the embedded decomposition algorithm based on the local least squares support vector to transform the pedestrian information into vectors, so as to realize the effective recognition of complex scenes. The research results of this paper can effectively improve the retrieval efficiency and improve the retrieval precision. On this basis, this paper intends to extend the above algorithms from the region level to the pixel level, reduce the influence of non-matching factors on model identification performance, and improve the efficiency of multi-task learning. Figure 9 shows the overall network structure.

In order to obtain higher quality candidate boxes, Li et al. adjusted NAE as the baseline network structure and proposed the SeqNet algorithm [31]. The main idea is to leverage the Faster R-CNN as a stronger RPN to provide fewer but more accurate and higher quality candidate boxes that provide more differentiated pedestrian feature embeddings. As shown in Figure 10, SeqNet consists of two serialized header networks that solve pedestrian detection and pedestrian re-recognition problems, respectively. The first standard Faster R-CNN headers were used to generate accurate candidate boxes. A second, unmodified baseline header is used to further fine-tune these candidate boxes and extract their identification characteristics.

The network structure of the above end-to-end target search model is built on the basis of Faster R-CNN. The pedestrian candidate frame generated by the RPN network is used to extract pedestrian features one by one from the feature maps in the trunk network. This network based on candidate frame can better locate pedestrians and extract high-quality pedestrian features. But, the dense candidate frame also leads to a lot of candidate frame coordinate calculation problems.

Therefore, Yan et al. proposed the first Feature-Aligned Person Search Network (AlignPS) without an anchor frame [32,33,34]. The AlignPS network directly abandons the RPN network. The baseline network is constructed by taking FCOS, a frameless network in the target field, and its structure is shown in Figure 11. Since there is no RPN network, there is no problem with calculating a large number of candidate boxes, but due to the absence of operations such as ROI-Align, the pedestrian feature embedment of AlignPS, an unanchored box model, must be learned directly from the feature map without explicit regional alignment. At the same time, in order to learn multi-scale features in order to adapt to the scale changes of the target, the AlignPS model uses the improved AFA of the feature pyramid FPN as a feature fusion module. In order to maintain the consistency of the same pedestrian features at different scales, the AFA network only outputs the feature map of the last layer as an input for subsequent tasks. In response to the misalignment of tasks, AlignPS converts the original detection priority to the re-identification priority and proposes the principle of the “Re-id priority”, which means that the output of the backbone network first imposes a pedestrian re-identification loss for supervisory training, and then the target feature is embedded into the input detector to complete the target detection task.

## 4. Attention Mechanism

The attention mechanism can perform weighted processing when processing input data strengthen important information attention and ignore unnecessary information when executing deep convolutional neural network tasks. Improved YOLOv8 adds ECA attention mechanisms to the VGG module of the backbone network to improve the accuracy and performance of the model by adjusting attention to input data.

### 4.1. Attention Mechanism of ECA Channel

Channel attention mechanisms have great potential to improve the performance of neural networks, but most of them are designed into complex structures to achieve better performance, which undoubtedly increases the complexity of the model. ECA attention module has a simple structure but can bring obvious gain effect. By analyzing the attention module of the SE channel, it is found that it is inefficient to obtain the correlation between all channels at the same time. Therefore, the ECA model is a further improvement of the SE model. Among them, the SE module uses a Fully Connected (FC) layer to represent channel interest, the ECA uses 1 × 1 convolution to represent channel interest, and uses one-dimensional convolution to realize information interaction among multiple channels, reducing the parameter values of each channel in the model. In each convolution process, only some channels play a certain role. In this way, the information exchange between different channels can be better realized. Based on the different channel scales, the size of the convolutional core can be flexibly adjusted to realize the mutual influence among the layers with a large number of channels, thus greatly improving the channel learning speed.

The ECA attention module is shown in Figure 12. Firstly, the spatial characteristics of the image are extracted, and the 1 × 1 × C image is obtained by using the whole average pooling method. On this basis, the 1D convolution model is used to solve the dimension of the 1D convolution function, and the channel interest of 1 × 1 × C is obtained, and the original input feature map is fused to obtain the image with the channel interest. The adaptive calculation formula for the convolution core size is
(10)$$k=\Phi \left(C\right)={\left|\frac{\log_{2}C+b}{\gamma }\right|}_{\mathrm{odd}}$$
where k represents the size of the convolution kernel; C indicates the number of channels. γ and b are the parameters of the adaptive calculation adjustment.$\;{\left|\cdot \right|}_{odd}:\;\mathrm{The}\;\mathrm{value}\;\mathrm{of}\;k\;\mathrm{is}\;\mathrm{odd}.$

### 4.2. eSE Attention Mechanism

The SE focus module dynamically adjusts the weights of each channel, thereby improving the overall performance of the system. In order to avoid the high modeling complexity, the first order FC uses the reduced parameter r to reduce the input characteristic channel C to C/r. Second order FC will reduce the channel length to the initial channel size C. During this period, the channel dimension decreases, and the channel information is lost. The eSE model further modifies the model, that is, there is only one FC level and only one C-level channel, from the original two FC levels to C level. Therefore, the robustness of the algorithm is improved on the premise that the channel information is not lost.

In the C2f module of YOLOv8, the input is first separated through the first convolution layer and is divided into two parts: one part directly passes through n Bottleneck, and the other part is separated again after each operation layer to create a jump connection. The multiple feature maps of all the branches are collected in the eSE module. As shown in Figure 13, after the eSE module is globally average pooled and fully connected, the feature map learns and outputs the channel attention information, and fuses it into the feature map in the form of elements through feature mapping.

## 5. Results

### 5.1. Experimental Data Set

The dataset used for evaluation is the widely adopted Cityscapes dataset, consisting of 5000 finely annotated urban street images. It captures complex urban traffic scenes, including varied lighting conditions, occlusions, and dense object distributions, making it suitable for testing vehicle and pedestrian detection algorithms. The dataset was divided into 2975 training samples, 500 validation samples, and 1525 testing samples. The diversity of the dataset enables comprehensive evaluation of our proposed algorithm in real-world conditions.

### 5.2. Experimental Environment and Parameter Configuration

Test environment: Ubuntu1804 OS, two NVIDIA TITAN Xp, Python3.8 training platform, PyTorch deep learning framework, YOLOv8s pattern library. Set the training parameters for this pattern to the Epoch size of 200 and the batch size of 32, and all other parameters are at default.

### 5.3. Evaluation Index

This paper mainly uses the following evaluation indexes to evaluate the results of the algorithm.

The calculation formulas for accuracy (P) and recall (R) are shown in Equations (11) and (12):(11)$$P=\frac{T_{\mathrm{TP}}}{T_{\mathrm{TP}}+F_{\mathrm{TP}}}$$
(12)$$R=\frac{T_{\mathrm{TP}}}{T_{\mathrm{TP}}+F_{\mathrm{FN}}}$$

The calculation formula of the average accuracy mAP is shown in Equation (13):(13)$$m=\frac{1}{N}\sum_{n=0}^{N}\int_{0}^{1}P_{n}\left(r\right)\mathrm{dr}$$
where N represents the number of categories of detection targets; P_n indicates the AP value of a certain category.

mAP@0.5 sets the IoU to 0.5, calculates the AP for each class of images, and finally averages all the classes, known as mAP.

Frame rate (FPS) represents the number of images a model can process per second, which is usually used to measure the real-time performance of the model, supplemented by the number of parameters and amount of computation (GFLOPs).

### 5.4. Experimental Results

Our proposed algorithm showed significant improvements in both vehicle and pedestrian detection compared to state-of-the-art models. The experimental results are shown in Table 1 below.

Our experimental results demonstrate that the proposed algorithm outperforms both the baseline YOLOv8 and several state-of-the-art detection models, including YOLOv5, YOLOv7, and Faster R-CNN. Our algorithm achieves notable improvements in detection accuracy for both vehicles and pedestrians. Specifically, our method achieves a vehicle recall of 97.19%, surpassing YOLOv8’s 94.31% and Faster R-CNN’s 92.52%. Additionally, pedestrian precision reaches 96.02%, a significant enhancement over YOLOv8’s 94.89% and Faster R-CNN’s 93.04%. These improvements can be attributed to the enhanced feature extraction and attention mechanisms that allow the model to more effectively capture complex scene dynamics. Moreover, the proposed algorithm achieves a mean average precision (mAP) of 95.23%, outperforming YOLOv8 (89.61%) and Faster R-CNN (90.89%). The enhanced performance of our model, especially in crowded scenes, underscores its robustness and applicability in real-world traffic environments.

### 5.5. Optimization of Loss Function

The best performances after the simplification and optimization of the original network model of YOLOv8 and the network model of this paper were respectively trained and tested, and comparative experiments were conducted. The experimental results are shown in Table 2 below. See Figure 14.

The table illustrates the performance improvements of various algorithms before and after the application of optimized loss functions. Our algorithm demonstrates the most significant gains, with the common test set AP increasing from 94.86% to 96.02%, and the intensive test set AP showing a dramatic rise from 71.44% to 90.3%. Despite the substantial improvements in accuracy, our model maintains a real-time detection speed, with the FPS slightly adjusting from 41.52 to 41.46. Compared to other algorithms such as YOLOv8 and SORT, our algorithm achieves a better balance between detection accuracy and speed.

However, despite the improvements, our algorithm faces challenges under conditions with extreme lighting variations or highly complex occlusions. Further research is needed to enhance robustness under these conditions and to reduce computational overhead for deployment on edge devices. Future work will focus on refining the attention mechanisms and expanding the model’s capability to handle a wider range of environmental factors.

## 6. Conclusions

The experiment verifies that the pedestrian and vehicle detection used in this paper is feasible, which not only improves the detection accuracy, but also meets the real-time requirements. The detection effect diagram of the final algorithm model in this paper is shown in Figure 15.

For vehicles and pedestrians in complex environments, this paper designs a new algorithm based on improved YOLOv8. In this project, an adaptive neural network model is added to the residual matrix to enhance its characterization of features, and a depth-separable convolution model is adopted to replace the traditional 3 × 3 convolution to reduce the parameters of the neural network. Experiments show that, compared to YOLOv8, this method has improved accuracy and other indexes. However, despite the improvements, our algorithm faces challenges under conditions with extreme lighting variations or highly complex occlusions. Further research is needed to enhance robustness under these conditions and to reduce computational overhead for deployment on edge devices. Future work will focus on refining the attention mechanisms and expanding the model’s capability to handle a wider range of environmental factors.

In conclusion, the algorithm proposed in this study provides substantial improvements in both detection accuracy and speed, making it highly applicable for intelligent transportation systems and autonomous driving technologies.

## Figures and Tables

**Figure 1 sensors-24-06116-f001:**
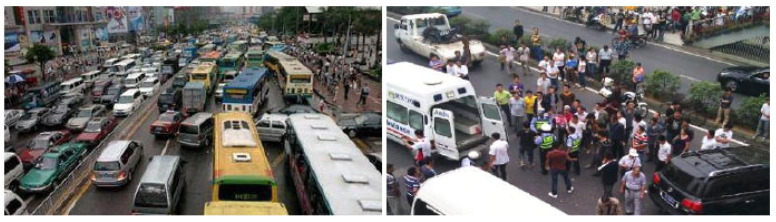
Traffic congestion and accident scene.

**Figure 2 sensors-24-06116-f002:**
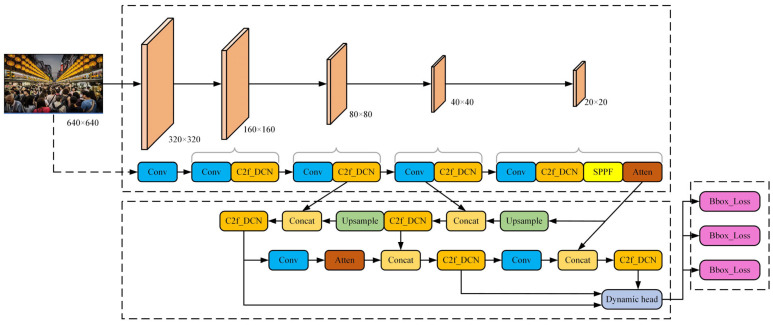
Network structure of improved YOLOv8 algorithm.

**Figure 3 sensors-24-06116-f003:**
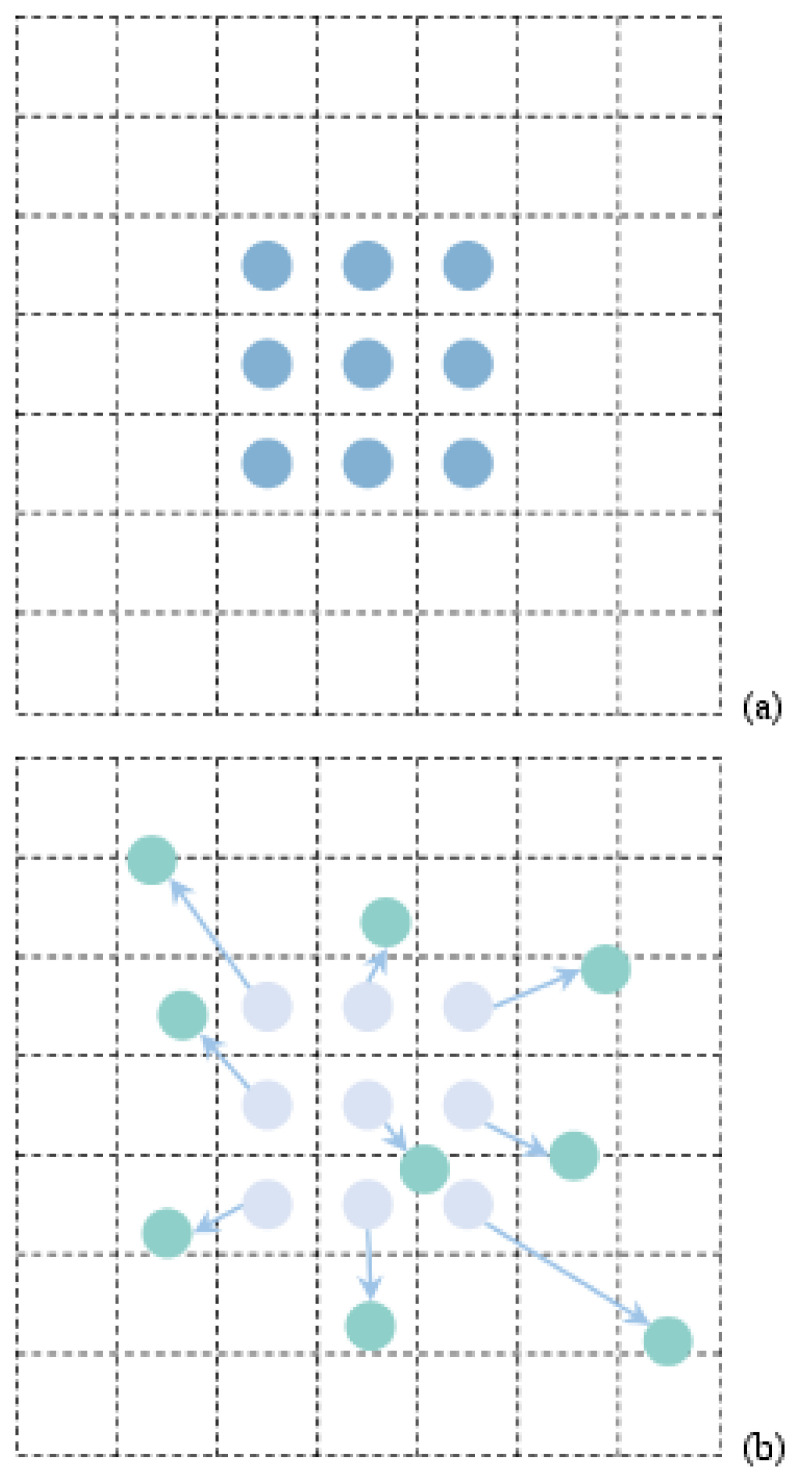
Sampling process ((**a**) Standard convolution sampling points; (**b**) deformable convolution sampling points).

**Figure 4 sensors-24-06116-f004:**
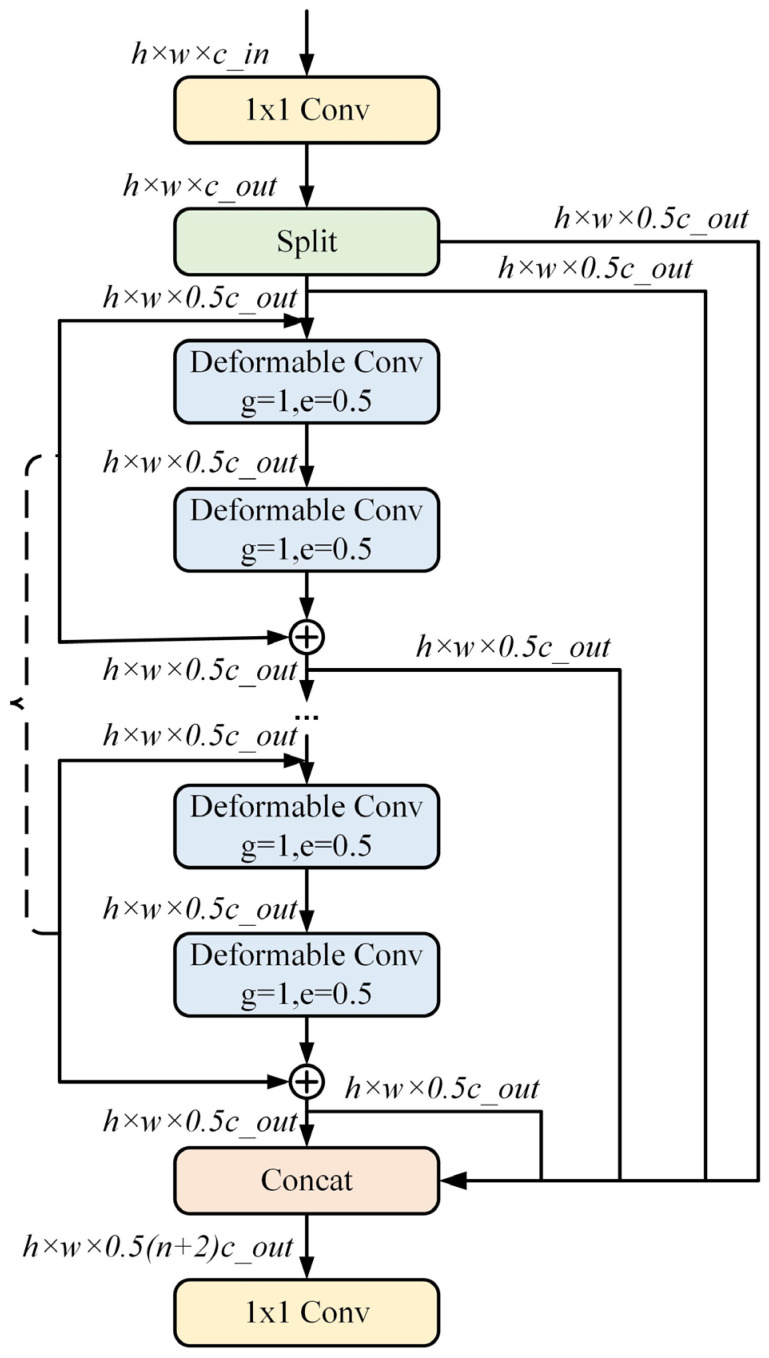
C2f_DCN module structure.

**Figure 5 sensors-24-06116-f005:**
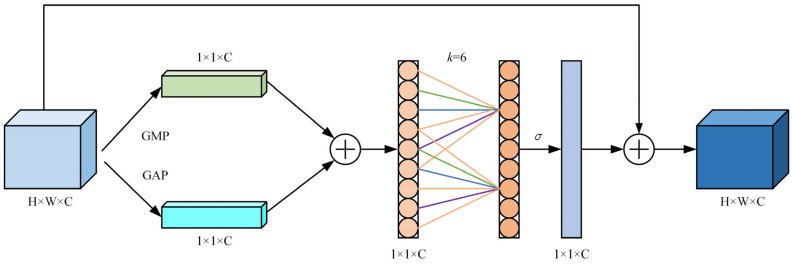
Occlusion perceptual attention mechanism.

**Figure 6 sensors-24-06116-f006:**
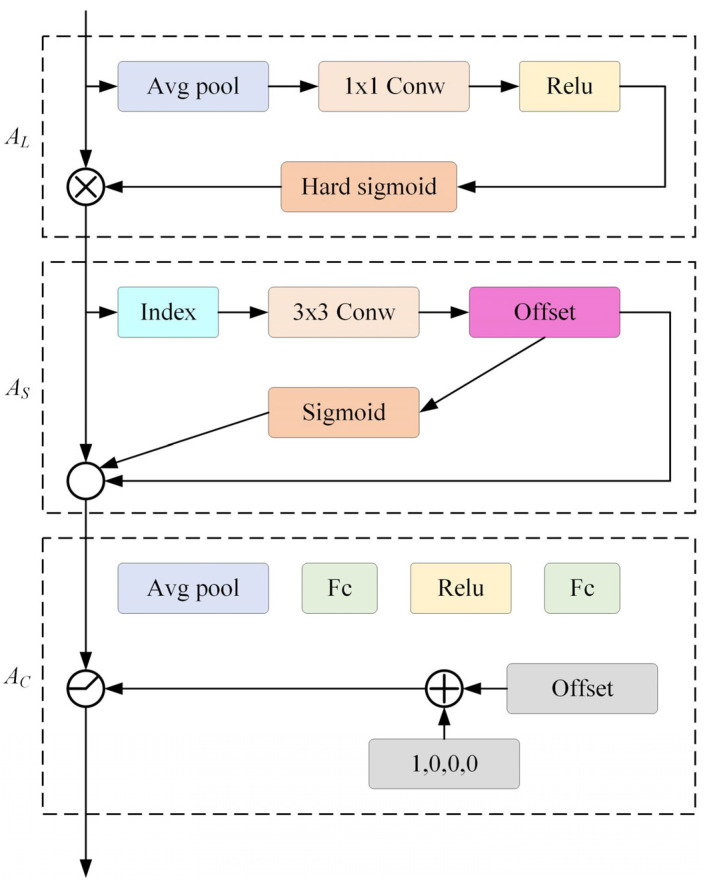
Structure of dynamic detection head.

**Figure 7 sensors-24-06116-f007:**
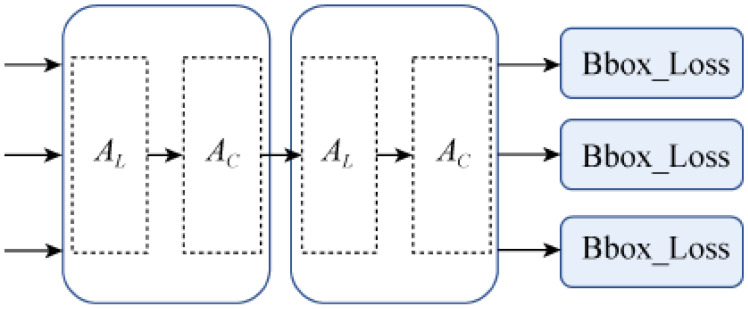
Dynamic decoupling header module.

**Figure 8 sensors-24-06116-f008:**
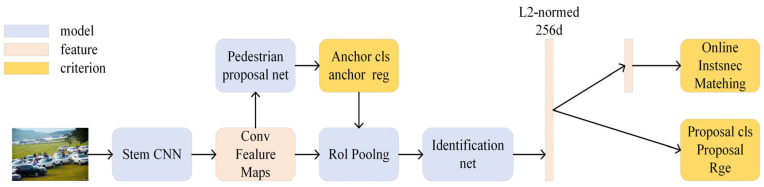
Schematic diagram of OIM structure.

**Figure 9 sensors-24-06116-f009:**
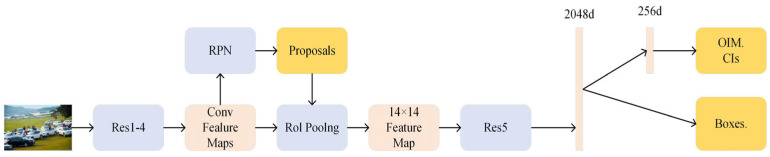
Schematic diagram of NAE structure.

**Figure 10 sensors-24-06116-f010:**
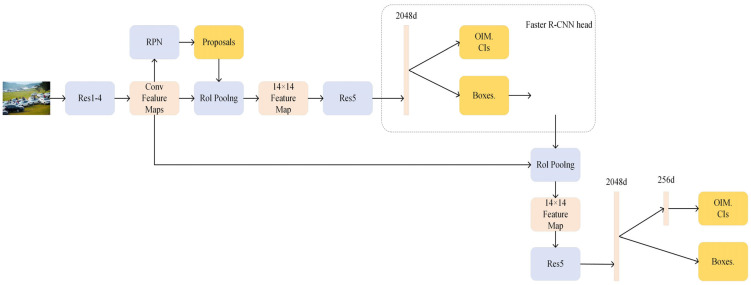
SeqNet structure diagram.

**Figure 11 sensors-24-06116-f011:**
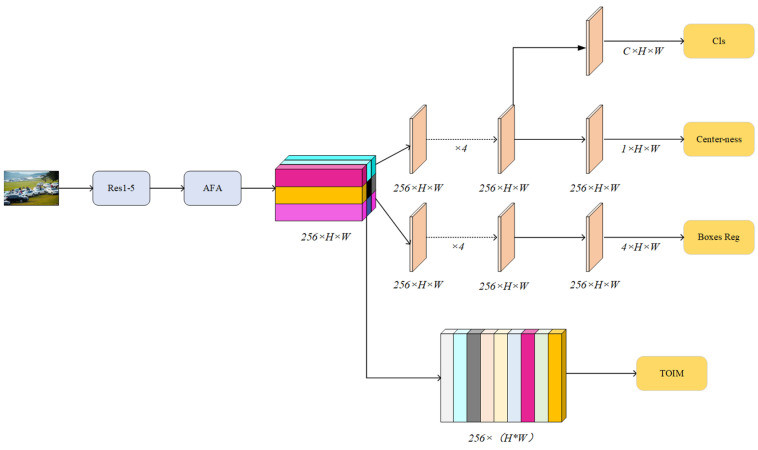
AlignPS structure diagram.

**Figure 12 sensors-24-06116-f012:**
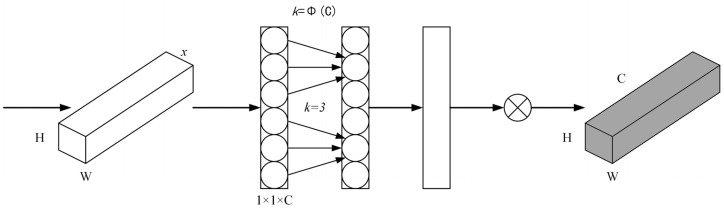
ECA attention module.

**Figure 13 sensors-24-06116-f013:**
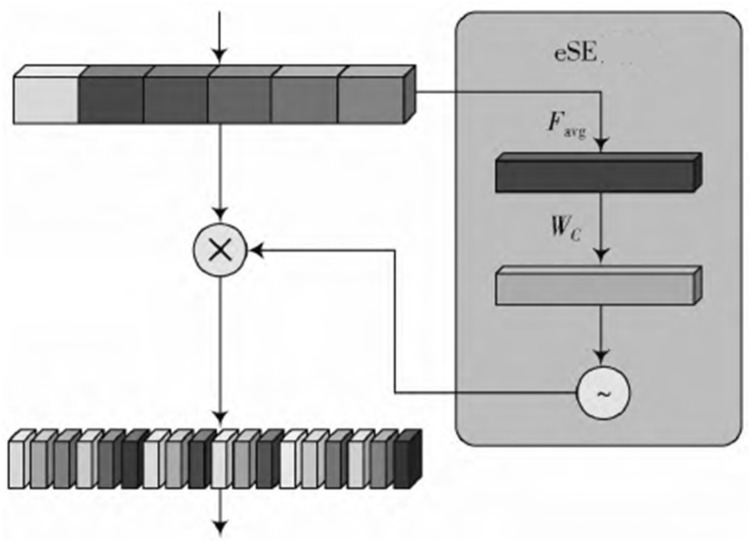
eSE module workflow.

**Figure 14 sensors-24-06116-f014:**
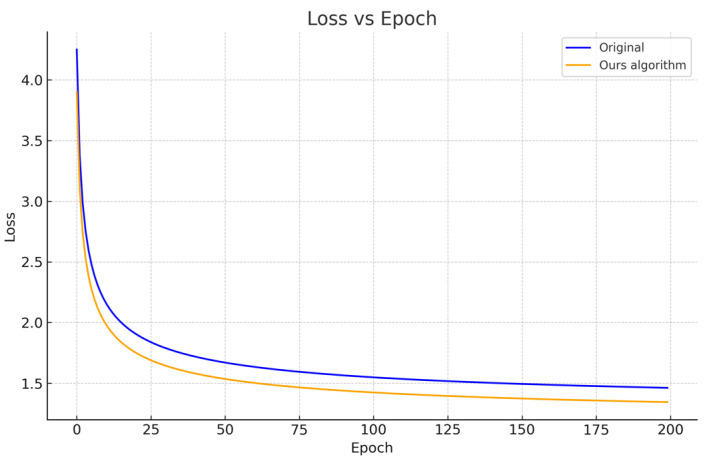
Comparison of training loss curves.

**Figure 15 sensors-24-06116-f015:**
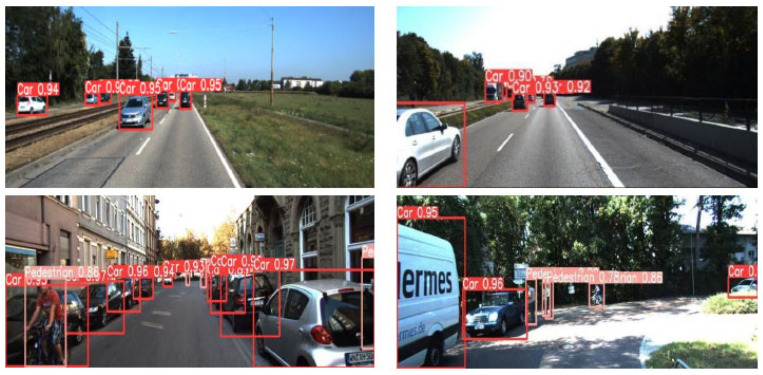
Model detection effect diagram.

**Table 1 sensors-24-06116-t001:** Comparison of algorithm test results.

Algorithm	Vehicle R/%	Pedestrian R/%	Vehicle P/%	Pedestrian P/%	Map/%
Faster R-CNN	92.52	88.13	88.47	93.04	90.89
YOLOv5	91.93	83.77	86.94	92.78	88.86
YOLOv7	93.44	84.28	87.06	93.33	89.23
SORT	91.57	82.47	85.13	92.32	87.08
YOLOv8	94.31	85.21	85.92	94.89	89.61
Our algorithm	97.19	88.65	96.02	98.84	95.23

**Table 2 sensors-24-06116-t002:** Experimental results before and after loss function optimization.

Algorithm	Loss Function	Training Duration (Days)	Common Test Set AP(%)	Intensive Test Set AP(%)	Detection Speed Fps
Faster R-CNN	before	4	92.81	71	15
	after	4.5	93.73	82	15.1
YOLOv5	before	3	91.52	68	40
	after	3.5	92.54	75	40.1
YOLOv7	before	3	92.11	69.5	45
	after	3.5	93.09	79	45.2
SORT	before	N/A	88.44	67.82	47.32
	after	N/A	89.57	77.67	47.39
YOLOv8	before	3	92.74	72	38.67
	after	3.5	93.86	84.2	38.71
Our algorithm	before	2	94.86	71.44	41.52
	after	2.5	96.02	90.3	41.46

## Data Availability

The original contributions presented in the study are included in the article, further inquiries can be directed to the corresponding author.

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
