# Peer review of "Synchronous End-to-End Vehicle Pedestrian Detection Algorithm Based on Improved YOLOv8 in Complex Scenarios"

_sensors, 2024, doi:10.3390/s24186116_

Round 1
Reviewer 1 Report
Comments and Suggestions for Authors
The paper emphasizes on proposed enhancements to the YOLOv8 algorithm for the detection of vehicles and pedestrians in intelligent transportation systems. While some sections of the paper are very well written and provide a good understanding of the research paper, there are areas which can be improved. While the study’s main objective and goals are somewhat clear, with interesting results being presented in terms of the specific case, there are certain parts that can be enhanced to improve the manuscript. The manuscript is generally clear, but there are sections where the sentence structure is overly complex, which may hinder readability. For instance, some sentences contain redundant information which can be reduced to make the wording more concise (e.g., the abstract could be shortened by removing repetitive statements). In addition, section headings need to be revised to follow the journal’s guidelines (see relevant author instructions) and some parts can be further improved. In general, the quality of the manuscript is solid and, if the authors are willing to revise their manuscript according to the comments below, I will be glad to review it again, as the research could prove highly beneficial, especially in road safety applications or even for use in AVs.
The authors are encouraged to revise their manuscript based on the comments provided in the supplementary Reviewer Report.

Overall, the quality of English in the manuscript is good, although it will still require further proofreading after the relevant revisions. There are occasional lapses into informal language that need to be addressed and revised (For instance, in Line 439, statements such as "has a great improvement" (Line 439) could be more formally phrased.
Reviewer 2 Report
Comments and Suggestions for Authors
Dear authors,
I will annotate the comments and questions as I read the manuscript.
As a general comment, the results and conclusions need to be extended. I was expecting that a random image would be presented in the results to showcase the comparison between the two models.
1. [Lines 45-46] "The economic losses caused by traffic accidents every year are terrible, which also highlights the urgency of traffic safety issues." What exactly you mean by every year are terrible? Can you, economicwise, add how the rate of economic loss has increased and incorporate a reference to support your claim.
2. [Lines 150-154] Change KTH for Kth.
3. [Figure 4] Improve the quality of the image, preferably in a vector graphic format. The same can apply to previous or posterior images where you believe the quality can be improved.
4. [Figure 8] Improve substantially this figure.
5. [Line 275, 295] The term "rerecognition" is made-up. I do not really understand what exactly you mean by re-recognition? Could you give me a brief answer?
6. [Lines 397-401] Can you reference the open dataset? What do you mean by "1525 samples"? Validation samples?
7. [Lines 403-406] It is not customary to write variables such as "batch_size" in the discussion. Correct it.
8. [Table 1] Rephrase "textual algorithm" to "proposed" or "ours". Moreover, can you comment on why the pedestrian accuracy is exactly the same, and moreover, the recall is also similar, why is that?
9. [Figure 14] Similar to all the images, please upgrade the quality. Also in the graph, attach a title for the x and y axes. Moreover, I believe this figure should not be in the conclusion.
10. The results are barely discussed. Can you also attach a random image where you compare the predictions made by your implementation and the base YOLOv8?
11. The conclusions are very vague. Please answer the following comments/questions and make sure these questions are reflected in the manuscript after modifying the conclusion section. Allow me to point out the several issues.
11.1) You are re-writing an introduction, e.g., "... the project intends to...", the reader by this point has read the manuscript and there is no point to introduce the aim of the manuscript.
11.2) You make a comment to your results, but nothing is remarkable. E.g., you say that there is a great improvement, I do not consider there is barely any improvement resultwise, in terms of architecture there is, but it is not even commented. About the improvements, there is no comment made regarding the fact that predictions of pedestrians are almost not improved, neither a discussion on why this could be. Also, the improvement on vehicles is minimum, of course the closest to 100% the more difficult it is to improve a model, but the accuracy of vehicles just improved from 89.92% to 90.02%. So the statement "has a great improvement in accuracy" is false. Then regarding "other aspects", that is not scientific sounding, please enumerate those other aspects (for example FPS) and backup those improvements with statements based on your results.
11.3) Which defects or limits your research has? Extend also this part of the conclusions and back it up with statements based on your results or methodology.
Comments on the Quality of English Language
1. [Line 142] "DAIT et al." I suppose It should be Dai. Check other instances of this reference and revise the reference list to avoid any other potential mistakes.
2. [Line 182] "..., In the same..." should be lowercase.
3. [Line 275] "rerecognition" is a made-up term.
4. [Line 299] I am not very sure if the word "project" or "this project" intends to refer to a real funded or non-funded project of yours, which should have been previously referenced somehow. Or by project you mean the manuscript under review. If it is the latter, please modify accordingly by article or paper and this also applies to the rest of multiple instances of "project", "this project", etc.
Reviewer 3 Report
Comments and Suggestions for Authors
Firstly, the abstract of the article does not highlight the prominent technical features of the paper, making it difficult for readers to grasp the main content of the paper from the author's abstract.
Secondly, the quality of the images in the article is very poor, which makes me question the originality of the images。
The article contains almost no tables and lacks ablation experiments, making it impossible to validate the author's work.
Moderate editing of English language required.
Reviewer 4 Report
Comments and Suggestions for Authors
1- The abstract did not indicate the standards adopted to demonstrate the efficiency of the proposed system, nor the values ​​obtained by the researcher. In addition to not clarifying the amendments used in building the proposed network.
2- This paragraph was not successful in specifying the limitations of the submitted work. Please explain it further:
At the algorithm level, the difficulty of this task lies in the feature extraction in complex crowd environment and the calculation of boundary frame loss in dense crowd. At the same time, because pedestrian detection algorithms are usually deployed on edge devices such as autonomous driving terminals and monitoring devices, higher requirements are also put forward to reduce the number of model parameters and the amount of computation and improve the real-time performance.
3- What is the difference between the Attention mechanism of ECA channel and other types of attention, and in which areas is it preferable to use it? This matter was not addressed
4-Figure 11 is not clear. Please improve its accuracy
5-Figure 12 needs redrawing
6-Add the link to the data set to the reference list.
7- The researcher did not explain the pre-processing of the data before sending it to the form
8- There is no discussion of the results, no conclusions for the work, or a proposal for future work in which the researcher seeks to work
9- Standardize the format of references
Reviewer 5 Report
Comments and Suggestions for Authors
This manuscript introduces an end-to-end target search algorithm to make the YOLOv8 algorithm more accurate in vehicle and pedestrian detection. In general, the results are presented not very clearly with inappropriate figures and tables. I recommend a rejection for publication.
1. too much backgrounds in the abstract and better to use more concrete numbers to address the results.
2. it is not clearly presented in the introduction why this manuscript is of significance.
3. it is not a proper section title for “2. Based on improved YOLOv8 detection algorithm”
4. figures should be modified with higher pdi of > 300.
5. results in section 5 is too weak to support the authors claim that the proposed algorithm is more stable and accurate in vehicle and pedestrian detection.
6. typos in line282 “Finally, use this data as the final feature extractor, as the final feature extractor.” , line 403 “Test environment: UUbuntu1804,”.
7. better to rewrite the conclusion section.
Round 2
Reviewer 1 Report
Comments and Suggestions for Authors
The authors hav addressed all of my comments. Only some minor revisions are necessary, after which I propose the manuscript for publication. The minor revisions are as follows:
Lines 57-62 there’s a repetition of the sentence “However, how to quickly and efficiently find…a major difficulty in current research.”
Lines 104-109: While the authors have addressed my comment for further providing context regarding different studies, there is a need to reference their comments in these lines with existing studies.
Lines 119-122: As above, this comment needs citing of existing studies in the literature, otherwise it is arbitrary.
Lines 233-244: Same as above, as mentioned in my first reviewer report, appropriate referencing to existing literature sources is necessary.
Reviewer 3 Report
Comments and Suggestions for Authors
Lack of Clear Motivation for Module Design: The summary does not clearly articulate the author's motivation for the module design from a technical perspective, making it difficult for readers to understand the underlying logic and purpose.
Excessive Length of the Research Background: The research background section occupies too much space, while the core technical contributions and innovations are underexplored, affecting the overall focus and effectiveness of the manuscript.
Lack of Comparison with Latest Methods: The comparative experiments do not include comparisons with the latest state-of-the-art (SOTA) methods, which limits the ability to demonstrate the relative advantages and contributions of the proposed approach.
Insufficient Persuasiveness of Experimental Design: The ablation experiments are only compared with the baseline, lacking sufficient data and analysis to convincingly demonstrate the effectiveness and superiority of the proposed method.
Insufficient Innovation: The manuscript lacks sufficient innovation, as the improvements to the network appear to be a combination of existing models rather than presenting a novel approach.
Overall Contribution is Not Sufficiently Highlighted: The overall contribution of the manuscript does not meet the journal's standards, particularly in terms of innovation and experimental validation, which need further enhancement.
Reviewer 4 Report
Comments and Suggestions for Authors
All comments done.
Author Response
Thank you very much for your time and valuable feedback during the review process. I am glad to hear that all comments have been addressed. Your suggestions have greatly improved the quality of the paper.
Reviewer 5 Report
Comments and Suggestions for Authors
I think the author answered my question well, and I suggest that this paper can be published after addressing two minor points.
1. better to add comparisons with other algorithms in table 1, such as faster-rcnn.
2. better to rewrite Conclusions for some logical problems, like the first paragraph could be further discussed in previous section (add more detailed discussion), and how "Therefore,..." comes?
